# Transcriptomic Approaches in Studies on and Applications of Chimeric Antigen Receptor T Cells

**DOI:** 10.3390/biomedicines11041107

**Published:** 2023-04-06

**Authors:** Karolina Pierzynowska, Lidia Gaffke, Jan M. Zaucha, Grzegorz Węgrzyn

**Affiliations:** 1Department of Molecular Biology, Faculty of Biology, University of Gdansk, Wita Stwosza 59, 80-308 Gdansk, Poland; karolina.pierzynowska@ug.edu.pl (K.P.); lidia.gaffke@ug.edu.pl (L.G.); 2Department of Hematology and Transplantology, Medical University of Gdansk, Smoluchowskiego 17, 80-214 Gdansk, Poland; jan.zaucha@gumed.edu.pl

**Keywords:** CAR-T cells, cancer, anti-cancer therapies, transcriptomics

## Abstract

Chimeric antigen receptor T (CAR-T) cells are specifically modified T cells which bear recombinant receptors, present at the cell surface and devoted to detect selected antigens of cancer cells, and due to the presence of transmembrane and activation domains, able to eliminate the latter ones. The use of CAR-T cells in anti-cancer therapies is a relatively novel approach, providing a powerful tool in the fight against cancer and bringing new hope for patients. However, despite huge possibilities and promising results of preclinical studies and clinical efficacy, there are various drawbacks to this therapy, including toxicity, possible relapses, restrictions to specific kinds of cancers, and others. Studies desiring to overcome these problems include various modern and advanced methods. One of them is transcriptomics, a set of techniques that analyze the abundance of all RNA transcripts present in the cell at certain moment and under certain conditions. The use of this method gives a global picture of the efficiency of expression of all genes, thus revealing the physiological state and regulatory processes occurring in the investigated cells. In this review, we summarize and discuss the use of transcriptomics in studies on and applications of CAR-T cells, especially in approaches focused on improved efficacy, reduced toxicity, new target cancers (like solid tumors), monitoring the treatment efficacy, developing novel analytical methods, and others.

## 1. Introduction

The development of anti-cancer therapies based on the use of chimeric antigen receptor T (CAR-T) cells can be recognized as one of recent breakthroughs in oncology, especially in the treatment of hematological malignancies [1]. CAR-T are specifically modified T cells which bear recombinant receptors, present at the cell surface and devoted to detecting selected antigens of cancer cells, and due to the presence of transmembrane and activation domains, able to eliminate the latter ones [1]. The construction, functions, and modes of action of CAR-T cells were reviewed recently [1,2], thus, we will present these aspects only briefly here.

CAR-T consists of three main domains which can differ in detail between specific constructs. The external domain (present at the cell surface) contains the single-chain variable fragment (scFv) of antibody, a construct that can specifically recognize selected antigens while being significantly simpler than the whole antibody molecule (for details on how scFv can be constructed and selected, e.g., using phage display technology, see the recent review article [3]; although scFv molecules are the most widely used antibody-like constructs present in CAR-T, different kinds of small alternative binding scaffolds, like Fab, nano-bodies, and others (reviewed in ref. [3]), can also been employed). The scFv module recognizing cancer-specific antigens (e.g., CD19 or BCMA) is connected to the linker (hinge) that joins the external domain to the transmembrane domain. This connection bridges the external domain and the internal domain, functioning as an activator of the cell. Such activation is mandatory to kill cancer cells recognized by the external domain (specifically, by scFv), thus, the internal domain contains not only a stimulatory module (like CD3ζ) but also co-stimulatory modules (like CD28 and/or 4-1BB) which enhance the reaction. The T cells bearing such engineered receptors are able to recognize specific antigens located at the surface of cancer cells irrespective of the presence/absence of major histocompatibility complex (MHC) molecules. After binding to the target cell, the CAR-T cells are activated to efficiently destroy the tumor [2].

Despite the successful introduction of therapies based on the use of CAR-T cells, there are various problems with this kind of treatment [2,4]. First, relapses have been observed after initial remission. Second, the toxicity of the CAR-T-based therapy was reported, consisting of mainly cytokine release syndrome (CRS) and immune cell associated-neurotoxicity (ICAN). Third, to date, only hematological malignancies can be treated using approved CAR-T therapies; therefore, broadening the cancer targets (especially to solid tumors) is highly desirable. Fourth, improvement of molecular monitoring of therapeutic efficacy is important. Fifth, new methods should also be developed to efficiently analyze the processes that occur during the treatment. Finally, basic studies on CAR-T cells are also mandatory to understand the cellular mechanisms happening in these special cells.

In the recently published review article leveraging various global methods for analyses of cellular states and functions, called “omics”, the improvement of CAR-T therapies was thoroughly discussed [4]. Indeed, perspectives on developing improvements in such therapeutic options are based largely on the use of advanced and sophisticated methods, as well as combinations of different kinds of omics, like genomics, transcriptomics, proteomics, metabolomics, microbiomics, and other specific types. Such multi-omics approaches can indicate detailed mechanisms by which CAR-T cells operate and act against cancer cells, especially when employing spatial multi-omics [4,5]. On the other hand, such complex methods are costly and only available in relatively few research centers, while in the case of specialized research on CAR-T cells, a single method can be sufficient and provide reliable results in economically reasonable conditions. Therefore, in this review, we will focus on the use of a single method (transcriptomics) in works on CAR-T cells and CAR-T-based therapies, and show the problem from a different perspective relative to the article above mentioned [4], namely what has already been done in this field, and how the obtained results may influence further investigations.

Transcriptomics is a set of techniques that analyze an abundance of all transcripts present in investigated cells under certain conditions. Since a detailed description of this technique has been summarized recently [4], we will only mention it here briefly. Early transcriptomic approaches involved the use of microarrays, based on large sets of molecular probes (oligonucleotides), specific for particular transcripts, mounted on a carrier [6]. This method made it possible to estimate levels of all cellular transcripts simultaneously; however, despite providing global pictures of cell physiology, the results were semi-quantitative at best (e.g., see ref. [7]). Therefore, microarrays are being replaced by more quantitative methods, especially those based on new-generation sequencing, like RNA-seq, giving significantly more precise results (e.g., see refs. [8,9,10]). Recently, a single-cell RNA-seq method has been developed and is used more and more frequently to indicate transcriptomic changes occurring at the level of a single cell, also providing information about divergencies and differentiation within a population of cells [11]. In this article, we will analyze published studies in which transcriptomic experiments were employed to analyze and improve various aspects of CAR-T-based therapies and discuss possibilities of further development in this field.

## 2. Methods

To collect articles published in the field of transcriptomic approaches in studies on CAR-T cells, the PubMed database (https://pubmed.ncbi.nlm.nih.gov; accessed on 24 February 2023) was searched using the terms “transcriptomic” and “CAR T”. A total number of 143 records was obtained on 24 February 2023. Preliminary analysis of these articles indicated the direct links between transcriptomics and CAR-T-related studies for 43 publications (other articles contained only some discussions, indirectly related to transcriptomics and/or CAR-T cells). These 43 papers were then further divided into sub-categories, describing results of studies and/or containing analyses/comments on (i) elucidating basic mechanisms of CAR-T functions; (ii) developing new methods of CAR-T analyses; (iii) improving the efficacy of CAR-T-based therapies; (iv) reducing the toxicity of CAR-T cells; (v) identifying new target diseases (like solid tumors) in the group of diseases that could be treated by CAR-T cells; and (vi) monitoring the efficacy of the treatment.

## 3. Transcriptomics in Elucidating Basic Mechanisms of CAR-T Functions

The first therapy based on the use of CAR-T cells was registered in 2017 (for a review, see ref. [2]), and before that date, no published studies on the use of transcriptomic methods in CAR-T investigations could be found. Thus, it is easy to conclude that transcriptomics has been employed in such studies for only six years or so.

Shortly after introducing the CAR-T-based therapies, the problem of different responses to the treatment by various patients was recognized; thus, it was mandatory to identify the mechanisms of processes leading to either therapeutic success or failure. In early studies in this field, the use of transcriptomic profiling indicated that in CAR-T cells derived from chronic lymphocytic leukemia patients responding effectively to the treatment, genes related to the immunological memory (like those coding for proteins from the IL-6/STAT3 group) were effectively expressed [12]. On the other hand, the cells of non-responders efficiently expressed genes coding for proteins involved in effector differentiation, glycolysis, exhaustion, and apoptosis [12]. These results demonstrated which processes can be responsible for the high and low efficacy of the therapy when stimulated due to the differential expression of specific genes. Quite similar studies were performed with the cells from four B-cell acute lymphoblastic leukemia patients treated with CAR-T cells, to demonstrate significant modulation of the expression of genes whose products are related to the cell cycle and immune response (especially the NK cell cytotoxicity and the NOD-like receptor signaling processes) [13].

An interesting study was conducted employing CAR-T cells from the products used for infusion and from blood samples of patients undergoing the therapy. When transcriptomic analysis was performed, it was demonstrated that clonal diversity of CAR-T cells was highest in the infusion products, while this feature was significantly less pronounced in samples from the blood of patients after infusion [14]. Therefore, the CAR-T cells of different transcriptional characteristics, present in the infusion product, had to be selected following infusion. Importantly, the selected clones of CAR-T cells revealed high-level expression of genes related to cytotoxicity and proliferation [14]. This might corroborate the results of previously mentioned studies [12,13] and indicate both transcriptional variability among CAR-T cells used for the therapy and selection of specific clones in the blood of patients. We suggest that the efficacy of the therapy may strongly depend on the kind of selected CAR-T cell clones after the infusion. In this view, a comparison of transcription profiles of CAR-T cells with different co-stimulatory domains (CD3ζ, 4-1BB-CD3ζ (BBζ) or CD28-CD3ζ (28ζ)) was fascinating, as it might indicate the most effective constructs. Such experiments indicated the gene expression signature common for CAR-T cells with the CD3ζ domain and those with the 4-1BB domain [15]. Moreover, it was shown that CAR-T cells with BBζ exhibited higher level expression of genes coding for human leukocyte antigens class II, ENPP2, and IL-21, while lower transcriptional activity of those encoding PD1, relative to CAR-T cells with 28ζ [15]. This kind of study on basic mechanisms of regulations occurring in CAR-T cells was also found to be important in testing the use of CAR-T cell in chronic myeloid leukemia, and ex vivo transcriptomic analysis suggested that this approach can be further used in developing personalized CAR-T cell therapy [16].

Further transcriptomic studies confirmed that there are dynamic changes in expression of various genes during the CAR-T therapy. Such an analysis was performed using transcriptomes of over 50 thousand cells, with the model of plasma cell leukemia, indicating characteristic changes in expression of genes coding for proteins related to cell proliferation, cytotoxicity, and intercellular signal transduction [17]. Moreover, it was shown that both tested types of CAR-T cells, bearing the scFv either against BCMA or CD19, had similar transcriptional profiles [17]. Another study employing a similar number of cells in the single-cell transcriptomic method revealed that upon exposure of the specific antigen (recognized by the CAR), transcriptional changes occurred in only about half of the population of CAR-T cells [18]. This may provide important information about the mechanism of poor response to the therapy in some patients.

When the process of exhaustion of CAR-T cells was studied, transcriptomic analyses led to identification of changes in expression of specific genes, including *PDCD1*, *CTLA4*, and *HAVCR2* [19]. Interestingly, significant differences in gene expression profiles during the T-cell exhaustion process were found between mouse and human cells [19], indicating the requirement of careful interpretation of the results obtained when using animal models in studies on gene expression regulation in CAR-T cells. To restrict the exhaustion process and to improve the efficacy of the CAR-T therapy, a novel type of CAR-T cells was developed which bears the CD79A/CD40 co-stimulatory module. To characterize this novel type of CAR-T, and to learn about the mechanisms regulating its activity, transcriptomic gene set enrichment analysis was conducted to demonstrate that elevated levels of expression were specific to genes related to cell proliferation, interferon signaling pathway, and naïve and memory T-cell signatures, while those connected to T-cell exhaustion were down-regulated [20]. Such studies were, therefore, useful in understanding the mechanisms of improved activity of CAR-T cells. Another approach to enhance the efficacy of the therapy involved overproduction of transmembrane amino acid transporters SLC7A5 and SLC7A11 which resulted in up-regulation of expression of genes coding for arginase I and arginase II, as confirmed by transcriptomic studies [21]. Such modifications of CAR-T cells stimulated their proliferation and anti-tumor activity [21].

In summary, transcriptomic studies which included profiling of gene expression, gene set enrichment analyses, and single-cell RNA-seq analyses, offered a powerful method to understand molecular mechanisms of functions of CAR-T cells in light of their anti-cancer activities. Moreover, they indicated processes which are involved in the exhaustion of these cells, which is the major cause of therapy failure and/or relapse of cancer. Finally, they revealed molecular bases of significant improvements of specifically modified CAR-T cells. Table 1 summarizes the processes affected by or in CAR-T cells, with examples of genes found to be up- or down-regulated because of transcriptomics.

## 4. Transcriptomics in Developing New Methods of CAR-T Analyses

Shortly after introducing the CAR-T therapy into clinical practice, attempts to improve the effectiveness of the procedures, reduce the toxicity, and efficiently monitor the course of the disease and the therapy appeared. These will be discussed in subsequent sections, however, to conduct such tasks, novel effective methods, considering global changes in investigated cells became required. The use of transcriptomic data offered the possibility to develop such methods, and in fact, several reports have been published in this field recently.

The first published attempt to use transcriptomics in developing new methods to improve CAR-T therapy was based on the task of identifying potential targets for CAR in acute myeloid leukemia [38]. The initially proposed procedure was based on the integration of large transcriptomic and proteomic sets of data obtained in experiments with both malignant and normal tissues. Using such data, an algorithm was developed which might allow researchers to identify potential target antigens which are present on the surface of leukemia cells, but not on normal cells [38]. However, this method failed to find potential targets with features as promising as CD19. A somewhat similar method was described a few years later, when a general procedure applicable in searching for CAR-T cell targets was proposed [39]. In this case, integrated data from transcriptomic and cell surface proteomic studies were used, presenting a method which might be applied not only to developing novel CAR-T therapies, but also to investigating infectious and autoimmune diseases [39].

One crucial point of transcriptomic studies is obtaining high-quality RNA samples. It is not an easy task, as RNA molecules are very unstable (due to nucleolytic degradation). This problem is especially pronounced in the work with T cells when it is also necessary to stain intracellular cytokines after permeabilization of cell membrane, exposing the RNA to degradation by enzymes. In this view, development of a protocol for high-quality RNA isolation from human T cells appeared as an especially useful method [40]. In this protocol, the specific steps consist of fixation with aldehyde, permeabilization with a detergent, staining of intracellular cytokines, and sorting. In addition, an RNase inhibitor and high-salt buffer must be used to prevent activities of nucleases [40]. It was proved that when using this method of RNA isolation, it is possible to obtain high-quality transcriptional profiles of functional subsets of T cells. Therefore, it was suggested that the described protocol can be useful in transcriptomic studies on CAR-T cells [40].

Single-cell RNA-seq transcriptomics is a powerful method to study the physiological state of the cell, however, analysis of transcriptomic data is complicated, especially in the case of T cells. To facilitate such analyses, reference T cell atlases have been developed, together with an algorithm for reference atlas projection, called ProjecTILs [41]. It was demonstrated that this algorithm accurately predicts effects of various cellular dysfunctions, and identifies alterations in expression of sets of genes under different conditions [41]. The single-cell RNA-seq data can be used not only to develop novel CAR-T therapies, but also combined therapies consisting of CAR-T cells, conjugated antibodies, and/or coated nanoparticles. When employing combinatorial optimization techniques (based on computational methods in which transcriptomic analyses are included), it was calculated that in the case of almost every cancer type, a combination of four therapeutic targets, considered in a personalized manner, should be sufficient to eliminate 80% of tumor cells, while saving 90% of non-tumor cells [42]. Then, using transcriptomic data, a machine learning model was developed which predicted the age of individual cells. This model was validated in studies on CAR-T cell expansion [43].

Apart from combined therapy, the use of standalone immunotherapy targets is another therapeutic approach. A scoring system, devoted to ranking potential targets, has been proposed in which proteomic and transcriptomic results were employed [44]. This approach led to the identification of several potential targets, like CCR10, TXNDC11, and LILRB4. To indicate the usefulness of this method, CAR-T cells recognizing CCR10 were constructed [44].

The new methods described above undoubtedly facilitated development of novel CAR-T therapies. The proposed procedures both provide high-quality material (RNA samples) for transcriptomic experiments and analyze transcriptomic data in order to improve CAR-T cells and/or identify novel targets. These novel methods are summarized in Table 2.

## 5. Transcriptomics in Improving the Efficacy of CAR-T Therapies

Improving the efficacy of CAR-T therapies is one of the major aims of current research on CAR-T cells. Construction of CAR-T cells producing hyper IL-6 (a designer cytokine activating the trans-signaling pathway: HIL-6) constitutively is one of the possibilities to enhance the anti-tumor activity. However, it was found that despite more efficient proliferation and improved anti-tumor efficacy, such modified HIL-6 CAR-T cells induced adverse effects, especially the symptoms resembling the graft-versus-host disease [45]. Nevertheless, transcriptomic analyses indicated that in these cells, expression of genes related to T-cell migration, early memory differentiation, and the IL-6/GP130/STAT3 signal transduction pathway are significantly enhanced. This encouraged the researchers to construct CAR-T cells producing a constitutively active GP130 protein. This construct revealed high anti-tumor activity while causing few adverse effects [45].

One of the major problems found in the CAR-T therapy is that only a fraction of CAR-T cells can effectively kill the target tumor cells. Therefore, single-cell transcriptomic profiling was employed to determine molecular processes which might be stimulated to improve the CAR-T-mediated killing [46]. This approach appeared effective, as CD137 has been identified as an inducible co-stimulatory protein whose gene is up-regulated in killer cells, but not in non-killer cells. Indeed, creation of CAR-T cells overexpressing a gene coding for CD137L proved that the anti-tumor activity of such a construct was significantly enhanced [46]. Another study aimed to overcome the resistance of cancer cells to EGFR-targeting CAR-T cells [47]. Transcriptomic analyses indicated that cancer cells resistant to killing by EGFR CAR-T cells are characterized by enhanced expression of a set of immunosuppressive genes. Therefore, a combined therapy with a specific CDK7 inhibitor (called THZ1) and EGFR CAR-T cells was tested to indicate that resistance, tumor growth, and metastasis were suppressed in the triple-negative breast cancer model [47]. Another work aiming to optimize the CAR-T cells used transcriptomic results to find that SOCS1 may serve as a major intracellular negative checkpoint of the adoptive T-cell response [48]. Such a discovery may open a new possibility to improve CAR-T therapy.

Transcriptomic studies were also employed to characterize the CAR-T cells constructed using the Solupore non-viral delivery system [49]. In contrast to CAR-T cells created using the electroporation method, which caused considerable changes in expression of genes encoding proteins involved in immunological reactions, the newly proposed transfection system did not cause such perturbations in the gene expression regulation [49]. Another method for modification of CAR-T cells is the use of the CRISPR/Cas system. The gain-of-function targets were identified, and it was shown that overexpression of the *PRODH2* gene (coding for proline dehydrogenase 2) can improve killing of various cancer cells by CAR-T cells [50]. The constructs were tested using transcriptomics which confirmed the assumed mechanism of action [50]. Not only enhancing expression of specific genes, but also the silencing of others may improve therapeutic properties of CAR-T cells. One example is miRNA-mediated down-regulation of expression of the *EBAG9* gene (encoding the estrogen receptor-binding fragment-associated antigen 9) [51]. The CAR-T cells with silenced *EBAG9* were more effective in cancer cell killing, while transcriptome profiling indicated a lack of genotoxicity or aberrant differentiation [51], confirming a good safety profile of such a construct. Finally, comparison of transcriptomes of CAT CAR-T and FMC63 CAR-T cells indicated that the former constructs revealed enhanced activation in response to CD19 stimulation [52].

The attempts described above to improve the efficacy of CAR-T-based therapies with the use of transcriptomics and specific modulation of gene expression are summarized in Table 3.

## 6. Transcriptomics in Assessing and Reducing the Toxicity of CAR-T Cells

The major safety issue related to the use of CAR-T cells is their toxicity, especially the most common complication—the cytokine release syndrome (CRS) [1,2]. Transcriptomic studies have been reported to be useful in both testing the levels of toxicity and determining its mechanisms which might help to prevent adverse effects, especially CRS [58].

A large-scale study, using the single-cell RNA-seq method, was performed to assess neurotoxicity of the CAR-T therapy of patients with B cell lymphomas [59]. Biological samples derived from 24 patients were investigated, with over 130,000 cells tested. This very interesting study identified transcriptomic patterns which might suggest molecular mechanisms related to neurotoxicity of the C19 CAR-T therapy and indicate possible ways to overcome this problem [59].

CAR-T cells can induce other organ toxicity. The assessment of the risk of toxicity to different organs caused by mesothelin CAR-T cells was performed using single-cell RNA-seq analysis [60]. That study indicated that myocardial, pulmonary, and stomach cells are of high risk of toxic effects caused by such CAR-T therapy. However, cells of esophagus, ileum, liver, kidney, and bladder expressed the *MSLN* gene (coding for mesothelin) at low levels, strongly suggesting that these organs are of low risk to be toxified by the tested CAR-T cells [60]. In another especially interesting article, a case report was presented where pulmonary flare-up developed after BCMA CAR-T therapy [61]. Single-cell RNA-seq study demonstrated that the gene expression profile identified in CAR-T-related pulmonary lesions was similar to those observed in sarcoidosis. It was also suggested that the CAR-T toxicity might be misinterpreted, suggesting the disease relapse, but nevertheless, the single-cell RNA-seq proved useful as a method suitable in making a reliable diagnosis [61].

The above review of the use of transcriptomics in assessment of toxicity of CAR-T cells indicates that this method provides a useful tool for identifying the mechanisms and levels of toxic effects, assessing specific risks, and facilitating adequate diagnosis. The examples of such studies are summarized in Table 4.

## 7. Transcriptomics in Identifying New Target Diseases for CAR-T Therapies

All previously registered CAR-T therapies are directed against hematological malignancies [1,2]. However, potentially each type of cancer should be treatable by specifically modified T cells. Thus, studies are being conducted to develop CAR-T therapies to treat various oncologic diseases, including solid tumors. Transcriptomic analyses may be especially useful in works devoted to introducing such therapeutic procedures, especially as early attempts to construct CAR-T cells that eradicate solid tumors were not fully successful due to limited efficacy.

One attempt to enhance this kind of CAR-T efficacy was to include co-stimulatory internal domains, MyD88 and CD40, into CARs [62]. Such constructs, called MC-CAR-T cells, revealed enhanced proliferative capacity and were more effective in killing cells of model solid tumors. To learn more about the molecular mechanism of the increased MC-CAR-T activity, transcriptomic experiments were conducted to demonstrate up-regulation of genes coding for MYB and FOXM1, which are crucial regulators of the cell cycle. Interestingly, following stimulation of MC-CAR-T cells, they expressed the *TBET* gene, coding for an important transcription factor, at the level lower than that observed in CD28 CAR-T and 4-1BB CAR-T cells, and thus, they remained in a less differentiated state [62]. Looking for the therapy of triple-negative breast cancer, it was found that MUC1-C is a crucial regulator of the transcriptome in this type of cancer [63]. Moreover, transcriptomic data indicated that gene expression induced by MUC1-C is mediated by STAT1 and IRF1 [63]. These results suggest that MUC1-C may be a potential target for CAR-T cells. Analogously, transcriptomic analyses were employed in studies focused on the development of a CAR-T therapy for pancreatic ductal adenocarcinoma [64]. A modified CAR-T therapy, revealing improved efficiency, was characterized by stimulation of expression of genes coding for granzymes, and impairment of expression of those encoding exhaustion markers [64].

It was demonstrated that that glioblastoma could be potentially treated with CAR-T cells specifically recognizing epidermal growth factor receptors (EGFR). However, frequent acquirement of resistance of glioblastoma cells to EGFR CAR-T cells is a severe limitation of this therapeutic method. To overcome this problem, transcriptomic analyses were performed to indicate that the resistance is correlated with enhanced expression of genes coding for immunosuppressive proteins [65]. The elevation of levels of transcripts of these genes was dependent on the presence of BRD4, a regulatory protein containing the bromodomain. Therefore, effects of specific inhibition of BRD4 were assessed to find that under such conditions, immunosuppressive genes were not activated [65], suggesting a specific way of improving the efficacy of CAR-T therapies against glioblastoma.

Analyses of transcriptomes were also a basis for identifying novel targets of C/G acute myeloid leukemia [66]. Namely, CAR-T cells directed against folate receptor α, encoded by the *FOLR1* gene and identified in transcriptomic studies as a potential new target, revealed high efficacy in vitro and in experiments with xenograft models [66]. Interestingly, other transcriptomic investigations led to the proposal of development of a cellular therapy alternative based on CAR-T cells [67]. Namely, NKG2C^+^CD8^+^ T cells were found as a lymphocyte population revealing low level expression of the *BCL11B* gene, coding for a transcription factor that regulates T cell developmental fate [67]. This feature has been proposed as attractive in light of developing anticancer therapies, especially against leukemia cells [67].

Examples of the use of transcriptomics in finding new target diseases (with new target genes) for CAR-T therapies are summarized in Table 5.

## 8. Transcriptomics in Monitoring the Efficacy of CAR-T Therapies

Monitoring therapeutical effectiveness using reliable methods is crucial for any type of treatment. The efficient monitoring of effects of CAR-T cells is also mandatory at the stage of the preparation of the therapeutic agents. The usefulness of transcriptomics in the latter approach has been demonstrated to assess the efficiency of delivery of a gene coding for CD19 CAR through lentiviral vectors [75]. Indeed, differential gene expression analysis has been reported as a method for unequivocal distinction between transduced and non-transduced cells during CAR-T cell construction [75]. At the other end of the therapeutical process is monitoring efficacy of the therapy in patients. Such studies were performed with 24 patients suffering from large B cell lymphomas. The results suggested that variation in efficacy and toxicity of CAR-T cells correlates with the level of molecular heterogeneity of the infusion products and that the molecular response at day 7 of the therapy might be used as a predictor for CAR-T therapeutic efficacy [76]. In fact, it was proposed that determination of gene expression signatures by single-cell RNA-seq might be used to monitor CAR-T cells at various stages of the therapeutic procedures, from characterization of the CAR design, through conditions of the production process and combination of therapies, to disease outcome [77]. It was even speculated that single-cell RNA-seq might be a standard method for monitoring of CAR-T cell therapies in the future [77]. Indeed, it was reported that analyses of efficiencies of gene expression might reflect a good correlation with patient outcome during CAR-T therapy [78].

Recent studies confirmed that the proposal of using transcriptomic analyses as primary means in monitoring the efficacy of CAR-T therapies is substantiated. Such analyses were employed to identify genomic signatures of clinical remission after treatment with CD19 CAR-T cells [79]. Patients with a complete response had transcriptomic profiles of bone marrow cells with efficient expression of genes involved in the T-cell activation, while non-responders revealed high levels of expression of genes coding for proteins involved in cell cycle checkpoint pathways [79]. During the phase I trial examining the efficiency of CD7 CAR-T cells in treatment of patients with relapsed or refractory T-cell acute lymphoblastic leukemia and T-cell lymphoblastic lymphoma, the single-cell RNA-seq method was used to profile the immune reconstitution [80]. Such analyses proposed putative relapse markers. Namely, expressions of *S100A8* and *S100A9* genes, coding for S100 calcium binding proteins A8 and A9, respectively, were significantly enhanced in leukocytes of relapsed patients relative to non-relapsed ones [80]. The S100A8/S100A9 oligomer is known for its role in innate immunity, as this protein sequesters metal ions outside the cells. Normally, this leads to starvation of pathogenic microbes for metal nutrients, thus inhibiting colonization of the host organism [81]. However, very recent studies indicated that S100A8/S100A9 plays a role in the cytoskeleton rearrangement through cross-linking F-actin and microtubules. This reaction requires the presence of calcium and the phosphorylation process [82]. Moreover, S100A8 and S100A9 are also ligands of the toll-like receptor 4 (TLR4) which is responsible for induction of the kinase cascade process causing the NF-κB pathway activation and then inflammation stimulation [83]. Therefore, we suspect that changed efficiency of expression of *S100A8* and *S100A9* genes might cause disturbance in the control of cell adhesion and motility and regulation of the inflammatory processes, influencing the ability to relapse, at least in the case of T-cell acute lymphoblastic leukemia and T-cell lymphoblastic lymphoma.

In summary, transcriptomic methods revealed a high potential in being the primary tools in monitoring various stages of development and the clinical use of CAR-T cells. These stages include designation of CARs, construction and production of CAR-T cells, and assessment of therapeutical outcome. It is, therefore, likely that transcriptomic analyses will be considered as primary means of monitoring the efficacy of CAR-T therapies in the near future.

## 9. Concluding Remarks

This review highlighted the high level of usefulness of transcriptomic methods in various aspects of studies and applications of CAR-T cells. These methods can be successfully employed in basic research on molecular mechanisms by which CAR-T cells act to kill cancer cells, in attempts to develop novel methods to assess different aspects of CAR-T cell biology, in works on the improvement of efficacy of CAR-T cells in anti-cancer therapies, in actions devoted to CAR-T toxicity assessment and reduction, in studies on indication of novel target diseases that can be treated with CAR-T cells (with solid tumors being the crucial targets for improved therapies, as discussed recently [84]), and in approaches to finding optimal methods of monitoring the efficacy of CAT-T therapies. The schematic summary of the use of transcriptomic methods in various aspects of the CAR-T cell technology is presented in Figure 1.

## Figures and Tables

**Figure 1 biomedicines-11-01107-f001:**
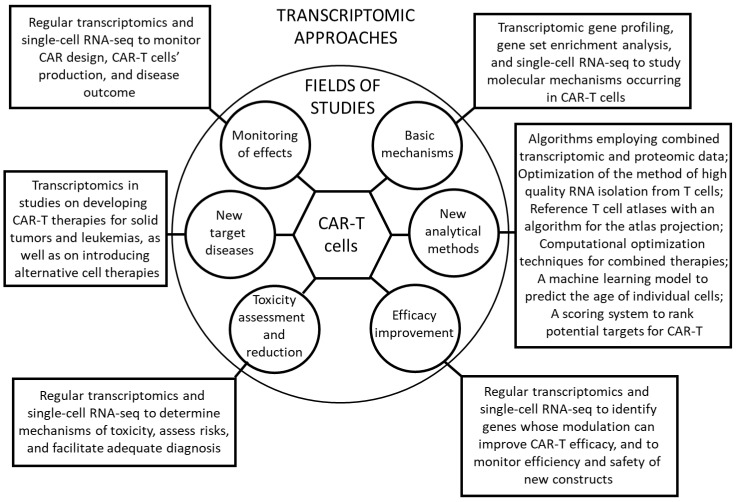
Schematic representation of the various possibilities of applications of transcriptomic methods in studies on and practical use of CAR-T cells, reported to date in the literature, as depicted in the form of the Cyske rosette chart (a kind of the schematic summary as proposed by Cyske et al. [85]).

**Table 1 biomedicines-11-01107-t001:** Processes modified by/in CAR-T cells as determined by transcriptomics.

Process Affected by or in CAR-T Cells	Examples of Genes with Changed Expression	Function(s) of the Gene(s) or Gene Product(s)
Immunological memory	*IL6*, *STAT3* [12]	Interleukin 6 (IL-6) is a proinflammatory cytokine; STAT3 is a transcription activator, stimulated in response to action of cytokines [22].
Cell cycle and immune response	*NRBP1*, *HIST1H4A*, *HIST2H4A* [13]	Nuclear receptor-binding protein 1 (NRBP1) is a tumor suppressor and pseudokinase [23]; *HIST1H4A*, *HIST2H4A* encode histone proteins [24].
Cytotoxicity and proliferation	*CCL4*, *NKG7* [14]	CCL4 is a chemokine [25]; NKG7 is a natural killer cell granule protein [26].
Human leukocyte antigens’ abundance	*ENPP2*, *IL21* [15]	Ectonucleotide pyrophosphatase/phosphodiesterase 2 (ENPP2) indirectly stimulates cell proliferation and chemotaxis [27]; interleukin 21 (IL-21) is a cytokine with immunoregulatory activity [28].
Cell proliferation, cytotoxicity, and signal transduction	*TNFRSF4*, *TUBA1B*, *RRM2*, *CCL3*, *CCL4*, *CCL5* [17]	TNFRSF4 is a member of the TNF-receptor superfamily which activates NF-κB [29]; *TUBA1B* codes for tubulin α1b [30]; CCL3, CCL4, and CCL5 are chemokines [25].
Exhaustion	*PDCD1, CTLA4, HAVCR2* [19]	Programmed cell death protein 1 (PDCD1) is an immune-inhibitory receptor [31]; cytotoxic T-lymphocyte-associated protein 4 (CTLA4) transmits an inhibitory signal to T cells [32]; HAVCR2 is a cell surface receptor with an inhibitory function [33].
Proliferation and interferon signaling pathway	*AIF1, HLA-DP, HLA-DR*, *HLA-DM, IFNGR2, GBP6, FLNB* [20]	Allograft inflammatory factor-1 (AIF1) is a protein that binds actin and calcium, and stimulates proliferation [34]; HLA-DP, HLA-DR, and HLA-DM are proteins from the major histocompatibility complex class II [35]; guanylate-binding protein 6 (GBP6) is induced by interferon and hydrolyzes GTP to both GDP and GMP [36]; filamin B (FLNB) is a protein connecting cell membrane to the actin cytoskeleton [37].

**Table 2 biomedicines-11-01107-t002:** New methods for transcriptomics in CAR-T studies.

Problem to Be Solved	Principle of the Method	Application(s)
Identification of potential targets for CAR	Algorithm based on transcriptomic and proteomic data [38,39]	Development of novel CAR-T therapies
Obtaining high-quality RNA samples	A procedure involving fixation with aldehyde, permeabilization with a detergent, staining of intracellular cytokines, and sorting, combined with the addition of an RNase inhibitor and high-salt buffer [40]	All transcriptomic approaches, especially for CAR-T studies
Performing efficient analysis of transcriptomic data	Reference T cell atlases with an algorithm for reference atlas projection (ProjecTILs) [41]; a machine learning model [43]	Prediction of effects of various cellular dysfunctions; prediction of the age of individual cells
Ranking of potential therapeutic targets	A scoring system based on proteomic and transcriptomic data	Identification potential targets for CAR-T therapies

**Table 3 biomedicines-11-01107-t003:** Improving the efficacy of CAR-T-based therapies using transcriptomics to identify gene(s) to be modulated.

Target Process(es)/Approach to Improve the Efficacy	Gene to Be Modulated	Function of the Gene or Gene Product
T-cell migration and early memory differentiation	*IL6ST* [45]	IL-6ST is also known as glycoprotein 130 (GP130) which acts as a transmembrane cytokine receptor involved in the IL-6/STAT3 signal transduction pathway [53].
Enhancement of tumor killing activity	*CD137L* (*TNFSF9*) [46]	CD137L is a ligand of the CD137 cytokine (from the tumor necrosis factor family) [54].
Overcoming the resistance of cancer cells to EGFR-targeting CAR-T cells	*CDK7* [47]	CDK7 is a cyclin-dependent kinase regulating the cell cycle progression [55].
Checkpoint of the adoptive T-cell response	*SOCS1* [48]	SOCS1 is a suppressor of cytokine signaling [56].
Estrogen-dependent signal transduction pathway	*EBAG9* [51]	EBAG9 is the estrogen receptor-binding fragment-associated antigen 9 which acts as a tumor-associated antigen, expressed with high efficiency in different kind of cancers [57].

**Table 4 biomedicines-11-01107-t004:** Transcriptomics as a method for assessment of toxicity of CAR-T cells.

Assessed Toxicity	Specific Method	Application
Neurotoxicity	Single-cell RNA-seq with over 130,000 cells tested [59]	Identification of transcriptomic patterns suggesting molecular mechanisms of neurotoxicity and indicating possible ways to reduce the toxic effects
Toxicity to different organs	Single-cell RNA-seq with samples derived from various organs [60]	Assessment of the risk for tested organs to be toxified by CAR-T cells
Pulmonary flare-up	Single-cell RNA-seq [61]	Making a reliable diagnosis and distinguishing between the disease relapse and sarcoidosis

**Table 5 biomedicines-11-01107-t005:** New target diseases for CAR-T therapies identified by using transcriptomics.

Target Disease	Target Gene	Function of the Target Gene or Gene Product
Glioma, B cell leukemia, osteosarcoma	*MYB*, *FOXM1*, *TBET* [62]	MYB is a proto-oncogene, a transcription factor involved in the control of hematopoiesis [68]; FOXM1 [69] and TBET (T-Bet) [70] are transcription factors involved in the regulation of cell proliferation and developmental processes, respectively.
Triple-negative breast cancer	*MUC1-C* [63]	MUC1-C is a protein from the mucin family, involved in protection of epithelia from the external environment [71].
Pancreatic ductal adenocarcinoma	*GZM* gene family [64]	Granzymes are serine proteases secreted by natural killer cells and cytotoxic T lymphocytes [72].
Glioblastoma	*BRD4* [65]	BRD4 (bromodomain containing 4 protein) is capable of chromatin binding and stimulation of transcription of genes involved in cell cycle progression, differentiation, and inflammation [73].
Acute myeloid leukemia	*FOLR1* [66]	The *FOLR1* gene codes for the folate receptor α that binds folic acid and transports 5-methyltetrahydrofolate into cells [74].

## Data Availability

Not applicable.

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
