# Peer review of "Transcriptomic Approaches in Studies on and Applications of Chimeric Antigen Receptor T Cells"

_biomedicines, 2023, doi:10.3390/biomedicines11041107_

Round 1

Reviewer 1 Report

In this manuscript entitled “Transcriptomic Approaches in Studies on and Applications of Chimeric Antigen Receptor T cells”, Pierzynowska, et al. concisely summarized the current status of CAR-T cell therapy and suggested a possible use of transcriptomics for the examination of the quality of immune effector cells. Since the effectiveness of CAR-T cell therapy is limited, it is imperative to establish the pre-evaluation method to predict the efficacy of immune effector cells used for adoptive immunotherapy. Transcriptomics is one of the strategies for the evaluation of the efficacy of CAR-T cells. In this manuscript, the authors selected 43 publications by searching Pubmed and divided them into six categories: (1) basic mechanisms of CAR-T cells, (2) novel methods for CAR-T cell analyses, (3) improvement of efficacy of CAR-T cell therapy, (4) reducing CAR-T cell toxicity, (5) identification of new target for CAR-T cell therapy, and (6) monitoring the efficacy of CAR-T cell therapy. Then, the authors systematically reviewed the role of transcriptomics in the prediction of the effectiveness and the future of CAR-T cell therapy, which allows for a better understanding of the control of the quality of immune effector cells for adoptive cell therapy.

Comments:

1.      The authors divided the transcriptomic analyses into 6 categories and discussed the details of them. It is good for readers to understand the role of transcriptomic analyses of CAR-T cells and to show the future direction of CAR-T cell therapy. However, just the description as a text is not reader-friendly. Whereas Figure 1 helps readers to grasp the whole idea of the authors, the inclusion of Tables in the six sections will help the readers to understand the context of the manuscript.

Minor comments:

1.      Page 1, Line 14: “brining” should read “bringing”.

2.      Page 2, Line 93: “CAR-T-base therapies” should read “CAR-T-based therapies”

3.      Page 3, Line 125: “the expression of genes which products are related to” should read “the expression of genes whose products are related to”.

4.      This manuscript should be edited by professional native speakers.

Reviewer 2 Report

The review highlights very important function of CAR-T cells such as transcriptional activity of genes. This can increase efficacy, decrease toxicity, find new markers associated with response to therapy, find markers associated with relapse. The review is good organized. I would suggest to present main genes or signatures, transcription factors associated with changes of CAR-T cell function with references. This will be more clear to reader.

Reviewer 3 Report

The manuscript presents a comprehensive overview of the insights into CAR-T therapies, delivered by transcriptomics and single-cell RNA transcriptomics, a valuable and timely topic. The importance of transcriptomics method for describing CAR-T therapeutic efficacy, as well as for monitoring and predicting their side-effects and prospective future development (such as applications for novel or less explored targets) are addressed. The review is well structured, the publications in the field are given sufficient credit, and its message is well outlined. The only remark from my side would be that when describing the outcomes of the studies, the authors should refer to the function (or potential function) of the relevant differentially regulated gene or group of genes involved, rather than just quoting their names, to elucidate the message of their comments. Please find below a list of minor remarks which I hope you will find helpful.

Page 1 lines 38-42: also other small alternative binding scaffolds apart from scFv can be used as a part of CAR - please broaden the statement.

Line 44: I propose using : this connection bridges, instead of using the description “domain” for the hinge region as well

Line 57: please spell out the ICAN abbreviation: Immune cell associated-neurotoxicity

Line 58: the options of treatment of solid tumors should be underlined, perhaps with a citation (https://doi.org/10.1002/jha2.356, or equivalent).

Line 93: based therapies

Line 106: in the group of diseases

Line 138: specific clones?

Line 139, and throughout the text: In this view … is a more common expression

Line 173: while those connected to?

Line 179: and their anti-tumor activity?

Line 237, and throughout the text: proteomics and transcriptomics

Line 243: RNA samples?

Line 266: EGFR-targeting CAR-T cells

Line 270: SOCS1, please briefly describe the function

Line 281: Not only

Lines 288-290: Please briefly describe the function of mentioned genes.

Line 315: please reword: but nevertheless, the single-cell RNA-seq proved useful as a method…

Line 385: genomic signatures and not genetic signatures

Line 390: Phase I trial examining the efficiency of CD7…

Line 394: please briefly describe S100 A8 and A9  derivatives (calprotectin) (maybe also their role as alarmins and TLR4 ligands).

Figure 1: Efficacy improvement, leading to: regular transcriptomics… I would recommend replacing “to test efficiency and safety” with “to monitor efficiency and safety”.
